# Facing the Human and Animal Brucellosis Conundrums: The Forgotten Lessons

**DOI:** 10.3390/microorganisms10050942

**Published:** 2022-04-30

**Authors:** Edgardo Moreno, José-María Blasco, Ignacio Moriyón

**Affiliations:** 1Tropical Disease Research Program, National University, Heredia 40104, Costa Rica; edgardo.moreno.robles@una.cr; 2CITA, IA2, Universidad de Zaragoza, 500569 Zaragoza, Spain; jblasco@unizar.es; 3Institute for Tropical Health and Department of Microbiology and Parasitology, Medical School, University of Navarra and IdISNA, 31008 Pamplona, Spain

**Keywords:** brucellosis, Malta fever, *Brucella*, diagnosis, vaccines, RB51, S19, Rev 1, DIVA

## Abstract

Brucellosis is a major zoonotic disease caused by *Brucella* species. Historically, the disease received over fifty names until it was recognized as a single entity, illustrating its protean manifestations and intricacies, traits that generated conundrums that have remained or re-emerged since they were first described. Here, we examine confusions concerning the clinical picture, serological diagnosis, and incidence of human brucellosis. We also discuss knowledge gaps and prevalent confusions about animal brucellosis, including brucellosis control strategies, the so-called confirmatory tests, and assumptions about the primary-binding assays and DNA detection methods. We describe how doubtfully characterized vaccines have failed to control brucellosis and emphasize how the requisites of controlled safety and protection experiments are generally overlooked. Finally, we briefly discuss the experience demonstrating that S19 remains the best cattle vaccine, while RB51 fails to validate its claimed properties (protection, differentiating infected and vaccinated animals (DIVA), and safety), offering a strong argument against its current widespread use. These conundrums show that knowledge dealing with brucellosis is lost, and previous experience is overlooked or misinterpreted, as illustrated in a significant number of misguided meta-analyses. In a global context of intensifying livestock breeding, such recurrent oversights threaten to increase the impact of brucellosis.

## 1. Introduction

In one of his books, the French novelist André Maurois (1885–1967) related that the Nobel laureate Iliá Metchnikoff (1845–1916), during his lectures at the Institute Pasteur in the early 1900s, used to show a map of the world, pointing out that Malta fever (brucellosis) was prevalent, precisely, in countries of the British Empire. Not without sarcasm, Metchnikoff lectured his students, saying that “this is not due to any evil influence of the British, but it merely means that they are the only people who have made studies on Malta fever and know how to diagnose it” [1].

Brucellosis is a zoonotic bacterial infection whose detection is not straightforward unless a person knows how to look for it, as was cleverly addressed by Metchnikoff. As first noted by Hughes in 1897 [2], the human infection courses have a variable, often long-lasting, incubation period, displaying a collection of non-pathognomonic symptoms (and/or signs), which are commonly confused with those of other maladies, even after causing death. In his seminal work on the discovery of *Brucella melitensis* (initially named *Micrococcus melitensis*) in 1887, Surgeon-Captain David Bruce noted that “microscopically the condition of the liver, spleen, and kidney was found to be very similar to what obtains in enteric fever, scarlet fever, and other micro-organismal diseases” [3]. Remarkably, in species such as dolphins, the disease commonly affects the reproductive and central nervous systems, causing abortions and death. In contrast, the infection in bovine, ovine, caprine, and swine remains silent most of the time, usually until the last portion of pregnancy, when abortions and increased perinatal mortality can become evident. Therefore, it is not unexpected that, more than thirteen decades since the discovery of *B. melitensis* and one hundred years of the description of the genus *Brucella* [3,4], the diagnosis, recognition, prevention, treatment, and management of the disease in humans and animals remains a puzzle. 

In 1912, only a few years before Alice Evans discovered the close relationship between *B. melitensis* and *B. abortus* affecting small ruminants and cattle, respectively, the Veterinary and Public Health services understood the link between animal and human infections through the consumption of raw dairy products [5]. Since then, the United States, Canada, Japan, Australia, New Zealand, several European countries, and a few others successfully eradicated brucellosis from domestic livestock and, consequently, from humans. However, ruminant brucellosis and the associated zoonotic cases remain prevalent in the rest of the world, mainly in middle-income and low-income countries and even in wealthy nations with structural deficiencies such as China, Russia, and some Latin-American and Arabic nations.

Unfortunately, a significant portion of the information, tools and accumulated experience regarding the control and eradication of brucellosis generated in the last century is currently neglected in light of the development of fashionable practices and methods. After high-income nations eradicated brucellosis from ruminants, the market for vaccines and diagnostic tools was considerably contracted in these countries. Consequently, “new” vaccines and diagnostic tests, which never became a relevant part of the tools used for eradication in wealthy countries, had to find other marketing options elsewhere. Not surprisingly, these “new” products are broadly commercialized in many low- and middle-income countries. Concurrently, and partly under pressure to publish meeting professional evaluations based on bibliometric parameters, an enormous volume of publications on the practical aspects of brucellosis have appeared in the last decades. 

The main purpose of this article is to discuss, in some detail, how many of these publications overlook solid previous evidence that remains essential to understanding how to tackle this zoonotic disease. We use this review to trace the origin of some misconceptions and clarify why they have become so common in some cases. To this end, it was necessary to describe the historical difficulties and most frequent problems confronted by veterinarians, medical practitioners, and brucellosis researchers. We first discuss the usefulness of the concepts commonly used to describe the variable clinical picture, the tools used for laboratory diagnosis, and data on the prevalence of human brucellosis, all of which e illustrate the difficulties intrinsic to this malady. Then, we consider several interconnected aspects of the animal disease, namely, the problems inherent in prevalence assessments, the value of the various serological and DNA-detecting tests, and the vaccines used or proposed for the control and eradication programs in several parts of the world. The evidence led us to vindicate the often neglected or misinterpreted bulk of the knowledge learned over more than thirteen decades, information that remains essential to combatting brucellosis. 

## 2. The Conundrums of Human Brucellosis

### 2.1. The Many Names of the Human Disease

Human brucellosis is a perplexing disease whose complexities are, at first glance, challenging to grasp. Foremost, in contrast to *Brucella* infections occurring in natural animal hosts, many symptoms and complications have been described in human brucellosis. After a variable, often long or exceedingly long, incubation period, the disease manifests inconsistently with diverse clinical signs, seldom associated with distinctive blood parameters or inflammatory markers (Figure 1) [6,7,8,9]. However, regardless of the protean clinical signs caused by the dominant zoonotic smooth (S) species (*Brucella melitensis*, *Brucella abortus*, and *Brucella suis*), the disease is treated through the same antibiotic regimen. Indeed, in addition to displaying a similar antibiotic sensitivity, they induce the same syndrome. In keeping with this complex picture, and despite the pathogen being identified at the end of the nineteenth century, it took over one hundred years to recognize the non-classical nature of a set of hidden virulence factors [10,11] whose common theme is a marked reduction in the structural details recognized by innate immunity [12]. This trait makes the brucellae stealthy pathogens of a broad range of vertebrates, thus multiplying their zoonotic potential [13].

Frequent outcomes of the indistinct picture of human brucellosis are delayed recognition, misdiagnosis, and underreporting. In the middle- and low-income countries where malaria, dengue, trypanosomiasis, zika, and chikungunya are endemic, human brucellosis is frequently mistaken for one of these febrile conditions [14,15]. This problem is compounded because, like a few other zoonoses, brucellosis has increased in many areas due to the intensification of animal breeding fueled by growing food demands [16,17]. Many immigrants and refugees from regions where the disease is highly prevalent travel to the handful of high-income nations that eradicated brucellae from domestic livestock. Consequently, human brucellosis is also an exotic infection or traveler’s disease, often misdiagnosed [18].

The difficulties in the clinical recognition of brucellosis became evident shortly after the causative agent of “Malta fever” was discovered by a team led by David Bruce in Malta. In a classical paper published in 1896, Surgeon-Captain Matthew L. Hughes gathered thirty-two names for brucellosis [19], and this list expanded as the disease was recognized outside the Mediterranean area (Table 1) [5,20,21,22]. In medicine, names are crucial in shaping the minds of those that deal with diseases. “Mediterranean fever”, used for many years, was stigmatic for the Mediterranean basin countries and overlapped with other febrile endemic diseases, such as malaria. Likewise, “Crimean fever” was reminiscent of the 1853–1856 war on the northern coast of the Black Sea. Although widely used, Undulant Fever describing the wavelike pyrexial curve depicted in textbooks (Figure 2), was abandoned because other infectious diseases coexisting with brucellosis can also display similar febrile patterns. Moreover, apyrexial periods of varying lengths often interrupt such waves, and a low-grade fever condition occurs in long-evolution cases typical of rural areas with no easy access to healthcare. There are, however, many other signs. Weldon Dalrymple-Champneys and Wesley W. Spink, in their classical monographs “*Brucella* infection and undulant fever in man” and “The Nature of brucellosis”, reported close to thirty-four signs and symptoms in hundreds of patients [7,23] and their observations were confirmed by many researchers and have been repeated in recent works [6,8] (Figure 1).

Ten decades have passed since Alice E. Evans, the famous American microbiologist, put an end to this labyrinth of names, suggesting the term “brucellosis” for a disease with many symptoms. She discovered a close relationship between bacteria responsible for “Malta fever” and “Bang’s disease” and proposed that they belong to the same genus. Following Evans’s ideas [20], in 1920, Meyer and Shaw joined these bacteria in the *Brucella* genus, honoring David Bruce [4]. Through the years, and after becoming infected herself in 1922, Alice Evans insisted on quoting the name “brucellosis” for the disease caused by members of the genus *Brucella*, regardless of the species [24]. She experienced the illness periodically for over ten years, keeping her from attending the meeting where she became elected as the first female president of the Society of American Bacteriologists (now the American Society for Microbiology). Fortunately, her teachings have lived on through generations of scientists working in brucellosis.

### 2.2. The Clinical Course of Brucellosis: A Journey from Chaos to Order to Chaos

During the first half of the 20^th^ century, there were attempts to define the different clinical types of brucellosis as an intermittent, ambulatory, undulatory, malignant, bacteremic, intermediary, and subclinical. These descriptions, however, did not satisfy most clinicians, who experienced difficulties in including the increasing number of brucellosis cases within one of these terms [25]. Spink categorized the disease according to its duration as acute if it lasted less than three months, subacute if it lasted for three to twelve months, and chronic if it lasted beyond one year, regardless of its severity [23]. Others, such as the French infectologist Marcel Janbon, avoided the acute or subacute designation stages and called them “septicemic” and “intermediary”. Still, he kept the chronic period designation for the last phase [26]. However, Spanish clinicians, overloaded with *B. melitensis* cases, stressed the difficulties of defining “chronic brucellosis” and observed that it seldom followed an “acute” state [9]. Some authors considered orchiepididymitis, spondylitis, arthralgias, and other complications that were accompanied or not by an intense and sudden febrile course as chronic manifestations, while others did not.

For most physicians, an acute infection means the sudden appearance of symptoms that last for less than two weeks with distinct blood markers, while a chronic condition worsens over an extended period. These profiles, however, seldom fit with the course of brucellosis and, depending on the author’s definition, each modality had (and still has) different meanings, which are not necessarily linked to its duration. Based on his broad experience, de Villafañe-Lastra recognized acute and chronic brucellosis as modalities related to the severity of the symptoms, no matter their duration [27]. The medical community broadly accepted this proposal [21]. On the one hand, “acute” referred to any form of brucellosis that suddenly manifested as a more or less violent febrile attack that called for immediate medical attention. The “acute” connotation included flare-ups accompanied by feverish symptoms regardless of whether they occurred weeks or months after apparent clinical recovery from a first attack. “Chronic”, on the other hand, implied a constant or intermittent state of illness or impaired functional capacity, with diffuse symptoms, which is difficult to test. Even after treatment, long-lasting brucellosis can develop into a “fatigue” syndrome with deteriorated health, including neuropsychiatric disorders. In the experience of Alice Evans, there was “a long delay before the correct diagnosis was given” that corresponded to “neurasthenia”, defined as “exhaustion, insomnia, irritability, and complaints of aches and pains for which no objective signs could be found” [24]. This condition arose in a fraction of cases that continued to present clinical manifestations after antibiotic treatment. It has been common to divide patients displaying focal diseases, such as spondylitis, arthritis and other tissue affectations, and those without such complications, who fit in the “neurasthenic” category.

Over 70 years ago, the eminent medical bacteriologist Maximiliano Ruiz-Castañeda, who attended over 5000 cases of human brucellosis in México, wrote: “to classify brucellosis as acute, sub-acute and chronic is not only difficult but even inconvenient” [21]. This statement implies that, as in other scenarios in science, in brucellosis, we have to observe the complexity, accept it and then build from the pieces we observe. The conundrum in defining the clinical course of brucellosis in these terms remains, and the “mélange” becomes evident in the vast bulk of published papers in which the stages of brucellosis followed personal criteria. Not surprisingly, these nomenclatures are a source of confusion. A meta-analysis on the association between brucellosis and cytokine gene polymorphism included 25 publications, despite the non-equivalent criteria used in those works to define “focal brucellosis” estimated as organ infections “characterized by the presence of symptoms or signs of continued infection for at least seven days” [28]. However, an examination of the analyzed papers shows that it is impossible to distinguish patients that displayed the so-called “focal brucellosis” (following the definition) from those that did not have *Brucella* in the organs. Various works do not describe how they discriminated between patients with “local” from “non-local” forms. Moreover, when *Brucella* isolation was attempted, it was mostly (if not always) from blood, and the diagnosis was presumptive in a significant number of patients making it impossible to know that the bacteria were localized in a specific organ. Predictably, the low agreement among the various studies reflected this defect in the analysis. Likewise, in a meta-analysis of the clinical manifestations of brucellosis, aiming to assess the so-called “disability weights”, the authors proposed an estimated score of 0.190 for “acute” processes and 0.150 for “chronic, localized brucellosis”; however, these terms had different meanings in the 57 revised studies; therefore, this DALY calculation is not useful [29]. Surprisingly, this meta-analysis disregarded most of the classical and relevant references that thoroughly described the clinical picture and frequency of symptoms and signs [7,21,22,23,24,25,26,27], including the significant differences in *B. melitensis*, *B. abortus* and *B. suis* human cases that necessitate an analysis that is contextualized according to the dominance of the corresponding animal hosts in different areas. Not surprisingly, the conclusions on the clinical manifestations and exposure risks in this meta-analysis do not add anything to those described many decades ago [7,21,22,23,24,25,26,27]. Another meta-analysis on osteoarticular brucellosis found a grouped prevalence near one-third, without a significant specific relation to the acute or chronic cases defined in the selected manuscripts [30]. Significantly, in one more meta-analysis concerning the changes in peripheral blood T cell subsets in patients with brucellosis, the authors stated that one of the main limitations in their study was the difficulty in unifying the meaning and staging of acute and chronic illness, which impeded achieving a more accurate analysis [31]. Even those who assert that the classical subdivisions of acute, subacute, and chronic are of somewhat limited clinical interest still use them to describe brucellosis cases [32]. This classification is not inconsequential: ignoring the complexities of brucellosis confuses and delays the recognition of a disease that, in addition to a thorough anamnesis and clinical examination, requires laboratory tests for a definite diagnosis. Nevertheless, even laboratory tests are part of the brucellosis conundrum.

### 2.3. The Repertoire of Diagnostic Tools and Their Misuse

The catalog of brucellosis diagnostic tools developed over 130 years is gargantuan, and it is challenging to find an infectious disease with more proposed assays than brucellosis (Figure 3). In general, the brucellosis diagnostic assays are divided into five distinct groups: bacterial isolation–identification, antigen detection, DNA detection, assessment of the antibody response, and assessment of cell-mediated immune response. While the first three are direct procedures, the last two are indirect methods. Despite this enormous diversity, the commonly used diagnostic assays can be counted with both hands.

Since this is a pathogen for which there is no latent carrier state in humans, the isolation and identification of *Brucella* organisms remain the gold standard in terms of diagnostic specificity (DSp). Several investigators have attempted to overcome the difficulties in the isolation of these bacteria through methods that directly detect the bacterium or their microbial components in tissues or blood (Figure 3); however, none of these methods have been proved to be more efficient and reliable than bacterial isolation. Although not of optimal diagnostic sensitivity (DSe), the chances of culturing *Brucella* increase when using well-established protocols [33]. The classical Ruiz-Castañeda biphasic method or modern incubators, both of which detect bacterial growth without the risk of repeated blind subculturing, circumvent most of the biosafety problems posed by these pathogens. Still, in this aspect, brucellosis is peculiar because of the prolonged incubation period (up to a month, depending on the method) needed before discarding a culture as negative. Therefore, a correct anamnesis and a rapid serological test are necessary to apply the correct antibiotics promptly, avoid the increase in relapses related to delayed treatment and not relax precautions to avoid laboratory infections. While field brucellae seldom, if ever, develop antibiotic resistance [34], culture is of paramount importance after suspicion of infection with the RB51 or Rev1 live animal vaccines because both can infect humans and they are resistant to rifampin and streptomycin, respectively, two drugs that are part of the best treatment regimens; besides, standard serological tests do not detect RB51 infections [35,36,37].

*Brucella* isolation is indispensable when an epidemiological inquiry on the species, biovar, or genovar identification is necessary. Nonetheless, experience shows that even the best protocols cannot show these bacteria in a fraction of human cases confirmed as brucellosis by other pieces of evidence. This limitation, linked to the low frequency and small numbers of Brucellae in the blood, intracellular location, and inaccessibility in some focal forms of the disease, is a difficult obstacle to overcome [21,33]. Nevertheless, suboptimal sensitivity is not the primary problem with this diagnostic tool. The sad fact is that brucellosis is typical of the less-favored world populations, and these hardly ever have access to bacteriological diagnosis. Much less experience exists in detecting *Brucella* DNA, which is obscured by the lack of consensus protocols in presumptive case definitions and puzzling observations on the possible permanence of *Brucella* DNA years after recovery [33,38]. In addition, the extensive implementation of these methods is hampered by inadequate infrastructure in many low-income countries. These circumstances, together with the convenience, safety, ease of use and sensitivity of some serological tests, explain why these tools remain a critical pillar of brucellosis diagnosis.

Since the 1897 seminal work by Wright and Smith on the differentiation of typhoid and brucellosis using the serum agglutination test (SAT) [39], the detection of antibodies has been invaluable. This test led Themistocles Zammit to the seminal discovery that goats were the primary source of human infections in Malta [5], the beginning of a reciprocal exchange between human and animal assays that have been both inspiring and a source of misconceptions concerning the so-called “confirmatory” assays (Section 3.5). In successive years, however, the method gave unreliable results because of the distinctive difficulties regarding brucellosis. These include the so-called blocking antibodies and prozone-like effects in SAT and the dominance of non-agglutinating antibodies in long-evolution cases, making titers in this test wane below the accepted SAT cut-off diagnostic titer(s). Although puzzling, blocking antibodies and “prozones” are of minor practical importance because both are rare and can be resolved using serum dilutions, as was recommended long ago in standard SAT protocols [40]. The detection of non-agglutinating antibodies, however, requires acid pH tests (Rose Bengal [RBT] and Brucellacapt), the *Brucella* Coomb’s test, or *Brucella* lipopolysaccharide (LPS) immunoassays with anti-IgG and IgA conjugates [33,41,42]. Therefore, in endemic areas, predominantly rural, where long-evolution cases are common, the sole use of SAT results in mis- and underdiagnosis and false seroprevalence estimates. Surprisingly enough, these peculiarities are overlooked in current brucellosis case definitions of prestigious centers, sometimes used as a reference in studies in endemic countries [43]. The USA Center for Diseases Control and Prevention (CDC) (https://ndc.services.cdc.gov/case-definitions/brucellosis-2010/; accessed on 1 December 2021) postulates that alternatively to bacterial culture, definitive evidence for brucellosis is “a fourfold or greater rise in *Brucella* antibody titer between acute- and convalescent-phase serum specimens obtained greater than or equal to 2 weeks apart”. This definition does not clarify the test, which is important when titers are given. In addition, many confirmed cases do not show such an increase because the immunoglobulins have often reached a plateau when the patients are tested or, for SAT, titers can decrease. Furthermore, titers persisting long after recovery are common depending on the test. The CDC presumptive evidence is a “*Brucella* total antibody titer of greater than or equal to 160 by standard tube agglutination test (SAT) or *Brucella* microagglutination test (BMAT) in one or more serum specimens obtained after the onset of symptoms”, which also fails to recognize the existence of non-agglutinating antibodies, or “Detection of *Brucella* DNA in a clinical specimen by PCR assay”. This serological evidence leaves out a proportion of true brucellosis cases, which, in all likelihood, are higher in rural endemic areas, and there is no consensus on PCR protocols (see above).

Another surprising stumbling block for many laboratories in endemic areas affects the interpretation of the serological reactions in terms of “antigen specificity.” As an example, in a recent serological study in Tanzania [44], the authors reported “acute” brucellosis caused by “*B. abortus”* and “*B. melitensis”* in 7% and 15% of children, respectively. More erroneous than the use of the “acute” category is the belief that serology can differentiate between *B. abortus* and *B. melitensis* infections, something in which this report is not an exception, as the authors found this idea in over 35 papers from 12 countries published since 1993. This interpretation, rooted in the A (*abortus*) and M (*melitensis*) antigenic scheme postulated in 1932 by Wilson and Miles [45], still common in textbooks, had already been disproved by the work of Ross in 1927 and was soon disproven by other investigators [46]. It also overlooks that a dominant distribution of the A and M epitopes is not species-specific in S brucellae, but depends on the biovar. Ninety-three years ago, Ross wrote: “The conclusion that can be drawn, both theoretically and practically, is that either *Br. melitensis* or *Br. abortus* can be expected to give a reliable [agglutination] result, in case of undulant fever, whether due to *Br. melitensis* or *Br. abortus* infection” [47]. In 1954, the Committee on Brucellosis of the National Research Council recommended using *B. abortus* suspensions to diagnose all presumptive human cases, regardless of the suspected infecting S species [40]. Immunochemical studies have shown that all S zoonotic brucellae carry a common dominant epitope overlapping a few structural subtleties that account for the diagnostically irrelevant A and M epitopes [48]. The incorrect identification of the infecting species based on faulty tests and supposed species specificity of the antibody response is harmful because it leads to mistaken epidemiological conclusions, but even more worrisome is the reason that *B. melitensis* and *B. abortus* suspensions can yield different results.

An oddity of these bacteria is their marked tendency to undergo dissociation from an S to rough (R) caused by a loss of the LPS O-polysaccharide (O-PS) facilitated by genetic peculiarities [49,50]. These mutations lead to the disappearance of the relevant diagnostic epitopes and provoke autoagglutination, presented as spurious titers and false positives in the absence of specific antibodies [48]. Thus, it is not only the “*abortus*” versus “*melitensis*” interpretation that is wrong, but also the conclusion that different “agglutination” titers with *B. abortus* and *B. melitensis* suspensions show *Brucella* spp.-specific antibodies when they actually prove inadequate antigen quality and standardization. Indeed, autoagglutination results in inconsistent proportions of S and R *Brucella* cells in the vials of the kits and in dispensing the antigens, low repeatability of the test and subjective interpretations of the results. These problems, traced to *B. abortus* and *B. melitensis* bacterial suspensions widely marketed as “febrile antigen” kits for human brucellosis, cause harm in low- and middle-income countries where physicians surprisingly place considerable faith in these tests. In a recent study carried out, looking at 887 cases, positive results in the “febrile antigen” test showed a weak or no correlation with most established risk factors for *Brucella* infection, with over 95% being false positives [51]. The consequences of subsequent misdiagnoses are dramatic. First, physicians wrongly administer the prolonged and expensive antibiotic treatment recommended for brucellosis, as was found in a recent study in Kenya [52]. Second, data on human brucellosis based on false-positive results hide the effect of vaccination of animals in reducing human disease (Section 2.4), a chief argument driving stakeholders to implement this very effective control measure.

Is human brucellosis serology so challenging? Whereas some could erroneously infer this from the array of proposed serological tests (Figure 3), the facts are that all currently used assays mostly, if not exclusively, detect antibodies to the O-PS and that simple protocols for the standardization and prevention of S to R dissociation were standardized in guidebooks a long time ago [53]. Since reference laboratories with the necessary reagents and skills for primary quality control (i.e., validation with a panel of well-characterized sera [41], a precaution that can also be implemented in routine laboratories using their collection of sera) are nonexistent in large areas of the world where the disease is endemic, the quality of commercial antigens is assumed. Regrettably, most makers do not clarify the validation procedures, if any, as exemplified by the case of the so-called “febrile antigens.” However, when proper precautions are taken, SAT with serum dilutions remains a handy diagnostic tool since it circumvents the blocking antibody and prozone issues, is affordable and is not technically demanding. Due to the intense agglutinating activity of IgM, its maximal usefulness is in areas with prompt access to hospitals; always keeping in mind that, even under these favorable conditions, SAT can yield false-negative results when disease incubation times are long [41]. Additionally, wherever access to health care facilities is limited, SAT has poor diagnostic performance because of the dominance of non-agglutinating antibodies in long-evolution cases.

Fortunately, brucellologists resolved this diagnostic issue in the 1960s. Using *Brucella* suspensions in an acidic buffer, such as the RBT, initially developed for cattle brucellosis [54], and the more recent *Brucellacapt* renders “non-agglutinating” antibodies agglutinating and removes the prozone-like effects [42,48,55]. Although both tests are highly valuable, the former is far cheaper, simpler, and faster. According to a recent study in Africa, the average cost of RBT/sample, including consumables, equipment, personnel, facilities, and quality control, is close to 0.69 USD [56]. Others have estimated the overall costs at 3.26 USD, including those associated with collecting samples [52]. Therefore, it is not surprising that RBT has remained the test of choice for diagnosing human brucellosis in resource-limited settings since it was first recommended in 1974 [42,57,58]. However, the literature often describes RBT as a “screening” assay of low DSp, requiring additional “confirmatory” tests [33]. This widespread idea originates in using “confirmatory” serological tests for animal brucellosis in vaccinated contexts (see Section 3.5). Indeed, as in all serological tests, RBT loses some DSp in endemic areas [41,59]. However, no serological test is “confirmatory”, and, unlike animal brucellosis, physicians also have invaluable anamnestic and clinical data to establish the diagnosis. In this context, some simple tests are “complementary” because they add additional information by assessing the antibody levels (the case of *Brucellacapt*) or identifying the IgM and IgG isotypes (lateral flow immunochromatography (LFA)), thus offering clues on the evolution of the infection. Nevertheless, all serological results should be contrasted with the clinical and epidemiological evidence and, if possible, with bacteriological findings. Not surprisingly, the RBT has been vindicated in recent systematic works, and simple modifications that improve its diagnostic value have been introduced [42,57]. A shocking fact hampering RBT’s adoption for the diagnosis of human brucellosis in low-income countries is that the test marketed for this purpose is over 50 times more expensive than the one marketed for animal diagnosis, when it is the same product (they can, in fact, be traced to the same original producer; J.M. Blasco and I. Moriyón, unpublished observations). More elaborate tests, such as the various indirect or competitive enzyme-linked immunoenzymatic assays (iELISAs and cELISAs, respectively; with each family of tests differing in conjugate specificity, detection system and, to a lesser extent, antigen quality), the fluorescence polarization assay (FPA), and others [33] do not outperform the more straightforward and uncomplicated tests. In addition, these more sophisticated tests are far more expensive and, unfortunately, few, if any, have been adequately validated for endemic areas. These “high-tech” methods were developed in countries with a suitable infrastructure when human brucellosis was over or in frank decline, and they require well-defined human serum samples for standardization and validation, which are seldom available in endemic areas or areas where clinicians lack familiarity with the many variables of the human disease.

### 2.4. Prevalence and the Reality of Numbers

The variable incubation time, long-lasting course, and lack of pathognomonic symptoms make brucellosis a frequently overlooked and underreported condition. Seventy years ago, James Steele, an officer from the Epidemic Intelligence Service of the United States, who was instrumental in developing the discipline of Veterinary Public Health, noted that the human brucellosis figures were quite problematic to reconcile due to difficulties in distinguishing new cases and relapses [60]. These issues are particularly problematic in endemic areas where brucellosis coexists with other febrile illnesses, obscuring the diagnosis, as was already observed by Bruce in 1889 [61]. Brucellosis is a zoonotic disease whose prevalence in humans depends on that of the domestic host, the livestock species infected, their density and movements, and the degree of implementation of the vaccination and pasteurization of dairy products, as has been shown throughout the history of the disease [60,62,63,64,65,66,67]. Among these variables, whether brucellosis affects bovines, small ruminants, and swine is of paramount importance because the preferred *Brucella* species for these hosts display different virulence levels, with *B. melitensis* being the most zoonotic species [21,66,68]. Thus, studies based exclusively on human cases in endemic areas outside the context of the prevalence of domestic animals, although useful, have limited value.

Misleading figures, maps, citations, “copy-paste” references, and phrases have become a common trait in brucellosis, not only in research papers but in seminars, proposals, leaflets, technical manuals, and many other circumstances. These problems need to be corrected for both practical and conceptual purposes. For seventy years, the World Health Organization (WHO) has published close to 1000 manuscripts related to brucellosis (https://www.who.int, accessed on 1 December 2021), with many of them describing its frequency in different parts of the world based on the numbers that countries report annually. The World Organization of Animal Health (OIE) also collects data provided by the local health authorities. However, when the figures of both sources are contrasted with field studies, it becomes evident that a significant proportion of the official data lacks precision. For example, while a randomized study in a rural at-risk population in the Huetar Norte Region of Costa Rica (170 inhabitants) estimated the human brucellosis prevalence at 12.5% (the year 2015), the Ministry of Health reported a total of 144 human brucellosis cases for the whole country (then with 4.5 million inhabitants), and only eight cases for the same northern region [69]. Similarly, discordant figures have been found in countries such as Colombia and México [70,71] and elsewhere [14]. In some cases, these incorrect figures may reflect the absence of tools, such as diagnostic tests and sustained epidemiological actions. In other cases, there is a lack of understanding regarding obtaining meaningful figures and how the various tests work under specific epidemiological conditions.

A chief example of misleading figures is the “500,000-brucellosis cases and proportion of underreporting cases” repeated throughout the years in sentences such as “about half a million human brucellosis cases are annually reported worldwide, but the estimated number of unreported cases due to the unspecific clinical symptoms of the disease is supposed to be 10 times higher” [72]. At least 14,500 entries can be found on Google by searching for different word combinations of the “500,000 brucellosis” sentence written in English and Spanish. It seems that the “500,000 brucellosis cases” was introduced into the scientific literature by Ruiz-Castañeda in 1946, who, in a specific context (see below), said: “In taking the actual numbers, we may guess that the true incidence of human brucellosis infection in Europe and America is between 250,000 and 500,000 cases per year. Impressive numbers for a disease that shows minimal registration of the incidence officially reported…” [21]. Before 1946, the authors have not discovered references describing these numbers. A review published in a prestigious medical journal in 2006 is frequently cited as a source of these figures [73], but the three references provided [32,74,75] do not depict information regarding the “500,000 brucellosis cases” or the “10 times higher” in the expected proportion. Corbel [74] clearly states: “The true incidence of human brucellosis is unknown. The reported incidence in endemic-disease areas varies widely, from <0.1 to >200 per 100,000 population”. The Joint FAO/WHO expert committee report of 1986 [75], also frequently cited by others as the source of the “500,000 brucellosis cases”, does not mention human incidences in any region or country (and these figures were not in any accessible WHO brucellosis report from 1960 to 1990). The third reference is a self-citation [32] that only includes a table showing figures for various countries in a given year (taken from the OIE and various National Health Ministries), which, altogether, barely reached 62,000 cases for 2003 (the last year reported). Strikingly, four meta-analyses published in 2017–2021 reported these numbers as a fact but, significantly, none credited the same sources [76,77,78,79]. It is worth mentioning that some WHO global estimates suggest a 340,000-19,500,000 range of human cases in 2010 [80] based on only 29 scattered studies judged as being of sufficient quality for data analysis [81]. This exceedingly high range illustrates the extent to which the incidence of the human disease is unknown and the difficulties in finding reliable data.

Seventy years ago, these difficulties were perceived by James Steele. In the mid-twentieth century, he devised a method to calculate the approximate number of cases in a given place and, from there, make projections for a broader region according to the epidemiological circumstances of each location [60]. For example, to estimate the incidence of human brucellosis in the United States, Steele first determined the difference between the cases in a given area estimated by serology and through medical institutions. Then, he distributed the re-estimated incidence in that area into groups according to occupation and living conditions (e.g., rural, semirural, urban), calculating the morbidity coefficient in each case and applying it to the total population. In this way, Steele calculated that the incidence United States was 10,909 cases, not the 4143 cases that were officially reported. Following Steele’s method, Mexican investigators estimated the human brucellosis incidence in 1914 cases in Mexico City during 1952 (then with about 10 million inhabitants) instead of the officially reported 257 cases [21]. The variables introduced in these assessments explain why Ruiz-Castañeda reasoned that the particular conditions of different latitudes prevented an extrapolation of this figure outside Mexico City, and even less so to other localities of the American Continent [21], a comment unknown to those that cite the 500,000 figure.

The figures on the epidemiology of brucellosis that are frequently cited in scientific journals or reports of countries or agencies require scientific proof. Certainly, some reports depict reliable values concerning human brucellosis prevalence or incidence in certain areas. However, endorsing data that have not been validated may be more harmful than accepting the straightforward fact that “we do not know”. Rephrasing Daniel J. Boorstin, the famous Educator and Librarian of the US Congress: “the greatest enemy of knowledge is not ignorance; it is an illusion of knowledge” [82]. The truth is that the prevalence of human brucellosis in many world areas remains elusive.

## 3. The Conundrums of Animal Brucellosis

### 3.1. True Gaps of Knowledge

As the cattle industry is prevalent worldwide, *B. abortus*-infected bovines have been the primary focus of attention. However, although sheep and goats (and *B. melitensis* infected cattle when this happens) are the most significant source of human contagions [21,83,84,85], brucellosis of small ruminants has historically been neglected because these animals are generally bred in resource-limited rural or remote areas. Still more neglected is the disease of water buffaloes, camels, and yacks, domestic animals of poor or nomadic human groups [86,87]. Owing to their comparatively marginal production abilities, brucellosis in these latter species has hardly been investigated and, despite the extended assumptions, there is no reliable information on the validity of serological tests or the safety and protective efficacy of current vaccines.

Another significant gap in the knowledge is brucellosis in wildlife. In addition to the natural domestic reservoirs, no wild species has proved to be a natural reservoir of *B. abortus* or *B. melitensis*. However, when the anthropogenic effects in wildlife management become intense, there is a significant upsurge in both types of infection, and some wildlife species can become a reservoir of these brucellae [88,89]. However, the impact of brucellosis in wildlife is always difficult to assess. Practically all serological tests applied in wild animals are the same as those used in domestic animals and, indeed, there is a consensus that most have not been adequately validated in wildlife species [14,79,90,91]. This gap is compounded by the fact that, for evident reasons, studies in most wildlife species imply convenience sampling and, if individuals are captured alive, it is seldom ethical to sacrifice them for necropsy and bacteriological analysis. Therefore, our understanding of brucellosis in wildlife is very limited, as evidenced in recent meta-analyses that, although applying elaborated statistics, acknowledge these problems [79,91]. Some authors hope that new diagnostic tools, specifically PCR directly applied to animal samples, will overcome most difficulties [79], but it is still premature to accept that this methodology immediately equates to bacteriological investigations (see Section 3.6). Nevertheless, although a complicated topic, this aspect of brucellosis is of great interest in the Ecohealth era.

### 3.2. Misestimating Prevalence

Non-industrialized countries are home to about 1.25 billion cattle and 1.9 billion small ruminants [48]. These figures, compounded with difficulties intrinsic to the accessibility and transhumance, and similar practices conditioned by the geography and climate, put the enormous challenge posed by any assessment of brucellosis prevalence where most of these animals reside into context. Not surprisingly, the prevalence and incidence remain mostly unknown in large areas of both hemispheres. True, there is an increasing number of publications from areas in which the disease is known to exist by indirect evidence, including abattoir studies and reports of human brucellosis in risk groups. Nevertheless, these studies often show a misunderstanding of both the diagnostic tools and the dynamics of the animal disease. Some studies rely on tests with lower sensitivity than recognized alternatives, such as competitive cELISAs or SAT [92], with the latter being affected by the same non-agglutinating antibody issues as the human disease [48]. Many focus on only one livestock species (usually cattle) in mixed-breeding systems, disregarding the significance of cross-infections, particularly *B. melitensis* in cattle [14]. Likewise, many cross-sectional studies are carried out following the “confirmatory test” strategy, usually combining the RBT for “screening” with either the complement fixation test (CFT) or iELISAs or cELISAs as “confirmatory” tests. As discussed below, and provided it is properly understood, this strategy is only helpful whenever vaccination with S strains (S19 or Rev1) is part of a correctly implemented eradication program [92]. Together with the extended use of quantitative tests that are assumed to be effective based on the “performance index” adjustment of DSe/DSp and/or not validated for the local conditions [92], this strategy results in imperfect seroprevalence figures. Furthermore, these studies are often designed to assess individual seroprevalence and not the herd/flock prevalence, a parameter of critical importance in brucellosis.

As described in textbooks written over 80 years ago [93], brucellosis in a given herd or flock can fluctuate between two extreme situations, from a suddenly noticeable presentation with high numbers of abortions and reproductive failures to a state in which, although these symptoms are hardly evident (and usually go unnoticed in extensive breeding systems), the infection is well established. Fluctuations can be caused by many variables related to management and, whereas the individual prevalence in an endemic and epidemiologically circumscribed area is often moderate, the proportion of infected herds, flocks, or farms usually remains high. Still, this high collective prevalence is an accurate indicator of the potential of brucellosis to flare up in conditions favoring transmission, including animal movements and intensification. Therefore, in epidemiological terms, brucellosis should be considered as a collective (herd, flock, etc.) infection. Illustrative of the extent of these problems, in a relatively recent review of brucellosis in Sub-Saharan Africa [14], 21 out of 46 examined publications published in the 2002–2017 period used the “confirmative” testing strategy, and only 20 reported the herd/flock seroprevalence. However, in many of the latter studies, the animals were selected randomly, regardless of age. Indeed, including the young ones (brucellosis usually affects animals of reproductive age) in a serological survey reduces the possibility of detecting infected herds/flocks with the low within-herd/flock prevalence characteristic of endemic areas.

### 3.3. From Wrong Prevalence and Official Eradication Programs to Reality

With a few exceptions, endemic countries agree that eradicating brucellosis is a primary objective. However, as can be found in recent reports, reviews or meta-analyses, homogenous data obtained with properly used tests are seldom available [94,95,96,97].

A scan of worldwide official programs reveals that testing and slaughtering (T/S) of the infected animals, with or without a herd/flock preventive vaccination, is the chief eradication strategy [98]. A first and extended problem is that the individual, instead of the collective prevalence, depicts the state of brucellosis in a country, area, or epidemiological situation. A second one is that eradication in resource-limited regions where individual prevalence is “low” (usually accepted to be below 2% with unclear basis [83,98,99]) can be achieved by the sole application of T/S, without suitable individual identification of all animals, control of movements and compensatory measures for diagnosis and culling [71]. Indeed, despite the widespread “sophisticated” eradication rules written on governmental documents, progress has been unsatisfactory wherever vaccination is insufficient (including the issue of ineffective vaccines discussed in Section 3.7) or nonexistent, and when owners have to cover the T/S costs [70,98,100,101,102,103].

Historically, the illusion that T/S alone can eradicate brucellosis under low-prevalence conditions has its origin in the successful experiences of Australia, Canada, the United States and some European countries. Phrases such as “the prevalence of brucellosis was reduced to such degree that all states and territories can now embark upon and continue with the final phase of eradication by test and slaughter” [104] have been accepted without considering the context. Undeniably, this statement only makes sense in regions that first attained a generalized low prevalence through adequate vaccination programs, combined with T/S and mandatory economic returns. Indeed, once vaccination was interrupted, these countries maintained strict individual identification, control of animal movements, and compensatory T/S actions [104,105]. Nevertheless, as a complement to the sentence quoted above, Godfrey Alton, a chief figure in the eradication of brucellosis in Australia, stressed that the “government has recently agreed to continue the funding of the campaign […], and provide funds to compensate owners for the slaughter of reacting cattle” [104]. In addition to economic compensations and compulsory vaccination, brucellosis eradication requires flawless individual animal identification, strict control of livestock movements and alliances among all involved stakeholders, as repeatedly noted [105,106,107,108,109,110]. As stated in the last sentence of the editors’ summary of the Brucellosis Symposium held at the University of Texas A&M in 1977, “the control and eradication of brucellosis is a joint effort between the producer, the veterinarian, and the laboratory” [107]. As demonstrated by the frustrating and deficient collaborative scenarios of many countries in which brucellosis remains a burden, control of brucellosis without these parties’ joint efforts is an exceedingly difficult endeavor.

Without these requirements and limited funds, a control program based on compulsory and sustained mass vaccination (of both young and adult animals) with an effective vaccine (Section 3.7) is the only feasible and most effective strategy. This strategy is valid even if the prevalence is reduced to a few herds/flocks, as it would be only a matter of time before brucellosis re-emerges if vaccination is insufficient or prematurely stopped [106,110]. Only when vaccination has been continued for at least two animal generations and a correct assessment (Section 3.2) of the herd/flock prevalence shows a reduction to shallow levels could eradication be considered. This strategy can be implemented through vaccination of young replacements plus T/S of adults (i.e., older than one year) and a diagnosis reducing the interferences in vaccinated animals. Still, these requisites have not been met or cannot be met in many scenarios. In 1981, David Berman, a University of Wisconsin professor whose thorough understanding of brucellosis made him, for many years, a reference in the field, wrote: “For most cattle populations in the tropics, particularly under traditional systems of husbandry, [S19] vaccination with or without serological surveys to furnish current information on prevalence, will be the control method of choice for the foreseeable future” [111]. At present, this remains true for many countries worldwide, where eradication campaigns designed on national or international drawing boards are unrealistic.

### 3.4. Sophisticated and Expensive Is Not Better

The collection of serological tests proposed for diagnosing animal brucellosis is pervasive (Figure 3 and [48]); however, as in human disease, relatively few are used. Indeed, the number of works assessing and re-assessing new and classical tests is very high, but few publications meet the standards necessary to obtain unbiased conclusions. As an example, a relatively recent analysis of works reporting DSe and DSp of tests for cattle brucellosis yielded 138 publications, of which 115 were not valid for DSe/DSe assessment, including several highly cited works and reviews that emphasize the superiority of indirect iELISAs, cELISAs or FPA [92]. The fact is that, when the evidence is rigorously examined, RBT, the equivalent Buffered Plate Agglutination test (BPAT), and some iELISAs (if properly validated) display similar and close to optimal DSe/DSp in non-vaccinated domestic ruminants [92,112,113]. Accordingly, these three tests (and possibly LFA; [114]) are suitable for the analysis of seroprevalence in the absence of vaccination (see Section 3.2 and, for vaccinated animals, Section 3.5). However, while RBT, BPAT, and LFA are simple homogenous qualitative assays that do not require precise adjustments, iELISAs, cELISAs and FPA cut-offs need fine-tuning for use in a given epidemiological situation and require specific equipment and adequate laboratory conditions [115] (see below). Nevertheless, simple qualitative tests are not equal: LFA requires species-specific IgG controls in the chromatographic strip and is more expensive than the multi-species RBT/BPAT.

The preceding considerations should not be construed to mean that newer assays are defective by themselves. Some iELISAs have been introduced for surveillance in some countries that eradicated domestic ruminant brucellosis, either combined with RBT or alone. However, it must be noted that none of these countries reached a brucellosis-free status based on iELISAs or other primary binding assays, tests that are “latecomers” in their programs. Other tests have not been significantly used in those countries (FPA) or are not sensitive enough (cELISAs). At present, iELISAs are primarily used for the surveillance of thousands of animals in *B. abortus-* and *B. melitensis*-free countries, as they are amenable to a robust automatization (not exempt of problems; see below), animal censuses and individual identification are available, and they are cost-efficient when personnel expenses in those countries are considered. However, these conditions are not met in the overwhelming majority of endemic countries.

Concerning throughput and labor costs, when discussing the use of RBT, the above-quoted Geoffrey Alton emphasized that “large numbers of tests can be done economically by hand” and that “ten million samples were tested in this way in Australia in 1979” [116]. Nevertheless, the use of RBT as a single test when no vaccination is implemented is hampered by the “confirmatory test” misinterpretation (Section 3.5). This conundrum is best exemplified by the extended use of cELISAs, whose imperfect sensitivity and cutoff adjustment issues [92] fuel the incorrect idea that RBT-positive results demonstrate a low DSp when a given cELISA is negative. There is hope that innovations in paper-based microfluidics could lead to a generation of straightforward, affordable, and easily stored brucellosis tests useful in complex settings; however, until then, RBT remains a reference due to its diagnostic performance, simplicity, and low cost [117].

Making simple things complicated is often driven by the logical belief that sophisticated tests, described by those working in countries where ruminant brucellosis is not relevant anymore and surveillance is a priority, are better. As a result, professionals and lawmakers in the brucellosis field now face the conundrum of neglecting simple and relatively cheap tests because they are not aware of their practicality, thus allocating resources to expensive, sophisticated ones. Instead, it would be more sensible to invest in local reference laboratories to harmonize, validate, and control tests, which are currently non-existent in most low-income countries.

### 3.5. The “Confirmatory Test” Confusion and the Pitfalls of Primary Binding Assays

It is well-known that the interpretation of serological tests is not straightforward in the context of vaccination plus T/S programs in infected areas (the typical situation). This event is due to the antibody response following S19 or Rev1 vaccination and the anamnestic responses of vaccinated animals to field *Brucella* strains. Indeed, the differentiation of healthy vaccinated and truly infected individuals is necessary to avoid unnecessary and costly culling [35,106]. Countries that eradicated brucellosis from domestic ruminants used Rev 1 or S19 vaccination of young replacements, combined with testing with RBT or BPAT in series with CFT. However, some RBT-positive–CFT-negative results correspond to infected animals that have not developed the isotypes and the antibody levels detected in the later assay. This fact makes it necessary to repeat CFT. These countries had an infrastructure that made it possible to quarantine and retest serially suspicious farms to investigate any increase in CFT titers caused by active infection or anamnestic responses to brucellae in the group of animals [92]. This protocol, which results in a relatively important and costly over-culling, is the origin of the “confirmatory test” misconception when this is an administrative “follow up” strategy requiring specific conditions and infrastructure. That is, in the real world, a “confirmatory test” capable of perfectly distinguishing *Brucella* infected from vaccinated animals does not exist.

Undoubtedly, CFT is a technically challenging test requiring strict and repeated controls and, as said, results in a lower DSe than RBT/BPAT [92,112,113]. Thus, although automation resolved most technical issues, primary binding assays have been adopted instead of CFT in many countries. However, overemphasizing the simplicity, robustness, and equivalence under different epidemiological conditions of those assays may be counterproductive. In addition to their higher price and sophisticated equipment, there are technical obstacles such as the degradation of conjugates, antibodies, antigens, control sera, and general laboratory conditions affecting the kinetics of the reactions. The response of the binding assays is non-linear; therefore, the cutoff adjustment depends on the internal controls, even in the same laboratory, a procedure that is not straightforward. In addition, their standardization requires well-categorized sera from infected animals (proved by bacterial isolation and not biased by previous serological analyses), from brucellosis-free herds/flocks, and correctly vaccinated animals, all representative of the epidemiological settings, breed and age groups. Obtaining sera is laborious, and many of the “confirmatory” primary binding assays have been optimized using biased control sera that logically result in biased DSe/DSp estimations [92].

Since laboratories in low-resource countries have meager possibilities of validating these primary binding tests to the local conditions, the cutoffs are used as suggested (sometimes in considerably large spans) in the manufacturer´s instructions, as a perusal of the literature shows. Illustrative of the relevance of these problems, an extensive trial in five Latin American countries [118] reported wide inter-laboratory differences in OD cutoff values of the same two iELISAs (70–16 and 14–73% of positive control serum) and cELISAs (18–35 and 18–44%), showing a high variability of the diagnostic parameters across laboratories and countries. Also, these kits were tested with sera that were previously selected by RBT and CFT from animals of unknown vaccination status [118]. These problems preclude obtaining suitable DSe/DSp estimates [92] and, therefore, any correct use in true epidemiological settings, as was subsequently found [101]. In some of these areas, reference laboratories supervising the quality and use of these tests are inoperative or inexistent. However, in other cases (mainly in middle-income countries), the laboratories are proficient, and the personnel are well-trained, but the countries´ legislation forces these techniques to use the recommended cut-offs [101].

### 3.6. Jumping to Conclusions: PCR as a Direct Diagnostic Test

Many investigators have developed PCR-based assays to characterize or detect *Brucella* DNA sequences. On the one hand, this methodology is a breakthrough regarding reliable identification, typing (classical biovar typing requires long experience and provides limited epidemiological information), and genotype analysis of *Brucella* strains isolated by conventional bacteriology. On the other hand, many of these assays have been applied to the direct detection of *Brucella* DNA in animal tissue or blood samples because they have the potential to provide superb DSe/DSp and simultaneous species identification. Since this would overcome both the inability of serological tests to identify the infecting species and some of the practical problems of bacteriology, it is a desirable possibility. Nevertheless, such an advance seems, at present, far away.

A PubMed search (Appendix A) reveals over 30 different PCR amplification protocols tested under laboratory conditions. Many of these in vitro studies report good analytical sensitivity/specificity, and some also report the detection of a few colonies in spiked samples, but less than a handful of them have been extended to test host animal samples. Moreover, when the specific protocols used in such animal samples (which include DNA extraction and purification methods, removal of inhibitors, possible different samples (dairy products, blood, vaginal fluids, tissues, organs and fetuses) and additional amplification and amplicon detection methods) are examined, the number of diagnostic PCR protocols is much higher.

This diversity poses a gargantuan problem in harmonizing and standardizing the PCR protocols if adopted for brucellosis diagnosis. However, the biggest issues are the proper assessment of DSe/DSp and any subsequent validation. The above-mentioned PubMed search identified 73 studies in domestic ruminants, camels, and pigs, of which 70 were not informative or valid regarding the DSe/DSp of the protocol used. Some report the results as “molecular evidence”, but most accept the results as proof of the presence/absence of viable *Brucella* commonly compared with imperfect bacteriology and/or non-validated or misused serological tests, or even without any other test as a reference. The DSp is particularly imperfectly assessed and/or assumed in the overwhelming majority of the studies, which leads to unexpected results. Some identified *B. abortus* S19 vaccine DNA in many semen samples of serologically negative bulls [119,120] or healthy seronegative calves before being vaccinated [121]. In another study, 148 serologically negative goats were found to positive by direct PCR [122], and similarly surprising results have been reported for sheep [123]. Likewise, based exclusively on a positive PCR, it has been concluded that “most […] camels are the noiseless transporters of *Brucella*” [124] and that *Brucella* is present in “apparently” healthy camels [125]. Obviously, there is no reason to accept a high frequency of *Brucella* infections in the absence of any clinical disease, with negative serology, no culture, and no follow-up investigations.

Last but not least, one study reported a Bayesian DSe/DSp estimate for the diagnosis of bovine, ovine, and caprine brucellosis. As often occurs when brucellosis tests are evaluated following this approach [92], priors were taken from unrelated studies (PCR of whole-blood and paraffin-embedded tissues of humans or canine brucellosis), or studies with no appropriate controls, and the prevalence used was the personal opinion of a local expert [126]. Notably, several studies concluded that the investigated PCR protocol did not result in a better DSe than the bacteriological culture (Appendix A).

Of the 73 studies, one in sheep showed close parallelism with RBT (a test of high DSe/DSp in these animals) and two are strictly valid for a DSe/DSp assessment of a given protocol, but only in concrete samples (cow milk; or semen and necropsy materials of *B. ovis* infected rams). Nevertheless, these two studies report that the DSe was not better than adequately performed bacteriological controls (Appendix A). Indeed, beating well-conducted bacteriology is a difficult challenge. Regarding DSp, the isolation of brucellae is the gold standard. Concerning sensitivity and threshold detection, *Brucella* cells can be isolated (using suitable methods and samples) from over 80 or even 90% of suspected individuals (for a brief discussion, see [92]), and culture can detect one colony-forming unit per g of tissue using protocols established many years ago [127,128]. Indeed, this is laborious and requires level 2 (not level 3, as often assumed) infrastructure, but both circumstances concur in DNA-detection methods, which also need costly equipment.

An additional unsolved question is to what extent DNA detection in animal samples equates to viable brucellae, an issue that is also raised by the use of PCR in human brucellosis (Section 2.3). There is a lack of information on *Brucella* DNA persistence in blood and tissues when the numbers of bacteria decrease even if the animals remain infected or when they are cleared, as in vaccinated animals. Naturally and experimentally infected animals can remain infected for months or even years but do not always show a positive serology [129,130,131]. Depending on the dose and route of administration, the live S19, RB51 and Rev 1 vaccines may persist for 1–3 months in 3–4 month-old vaccinated animals and for much more extended periods in adult vaccinated animals [37,130,132,133,134,135]. In all these cases, whether an optimized detection of *Brucella* DNA matches the presence of viable bacteria or DNA remains for longer in bacterial debris remains to be investigated. This event is critical if this methodology is applied in infected areas where vaccines are used.

At present, taking it for granted that PCR in any of its variants is a reliable tool for direct diagnosis is a premature conclusion and further studies are necessary. Consequently, the OIE has not approved and recommended any direct PCR technique to diagnose, control or eradicate brucellosis or estimate its prevalence in animals [136]. Indeed, there is hope that the problems and gaps outlined above will be overcome, including implementing simpler and less equipment-demanding methods. Nevertheless, these achievements will depend on a thorough understanding of the classical serological and bacteriological diagnostic methods that are necessarily used as a reference, on an awareness of the issues related to the dynamics of *Brucella* strains in naturally infected and vaccinated animals and, as a first step, on the use of samples that fulfill the necessary criteria for a proper assessment of any brucellosis diagnostic test [92].

### 3.7. Ariadne’s Thread in the Cattle Vaccine Labyrinth

Although the few countries that have eradicated *B. abortus* and *B. melitensis* from domestic ruminants used just two animal vaccines, the menu is more extensive. It includes a sizable set of vaccines, including live DIVA (i.e., with properties that make it possible to differentiate infected and vaccinated animals) ones and subcellular constructs, which are all claimed to be safe and/or effective. However, are they all safe and useful? This topic is of the utmost importance, considering the widespread poor understanding of most aspects of this neglected disease and the commercial (and perhaps also geopolitical) interests involved.

Brucellosis vaccine evaluation is a lengthy, expensive and complicated process in which controlled experiments are only a first but very valuable step to assess protection and safety (no milk excretion and/or vaccine-induced abortions in pregnant animals) [35,137]. Still, the quality of controlled studies depends on several experimental requisites that are not easy to implement (Table 2) and, unfortunately, a good understanding of the methodology established many years ago by workers with extensive experience on animal brucellosis is vanishing. This unfortunate event is shown in recent publications that claim the safety and/or efficacy in cattle of influenza virus vectors carrying genes of *Brucella* L7/L12 or Omp16 [138,139,140], a mutant (wrongly characterized as deleted in the perosamine synthetase gene) in buffaloes [141] and a 16MΔ*vjbR* mutant in small ruminants [142,143]. Likewise, some recent meta-analyses of controlled experiments use publications that do not meet the requisites in Table 2 and apply elaborated statistics to data that are not amenable to quantification and statistical comparisons because of their qualitative character. Through this methodology, inactivated vaccines are superior to the live reference vaccines [144], a conclusion that disregards why these vaccines were discarded many years ago [35,145]. Other meta-analyses “suggest that the dose of 10^9^ CFU for S19 and 10^10^ CFU for RB51 are the most suitable for the prevention of abortion and infection …” and that the study “provides very relevant information for brucellosis control and eradication … that can drive adjustments in vaccination schemes and brucellosis control modelling” [146]. Nevertheless, the “prevention of abortion” by itself (i.e., without bacteriological results proving the absence of brucellae) is counterproductive (Table 2). Moreover, these conclusions overlook both the results obtained with RB51 (see below) and the solid body of knowledge derived from the experience of countries that successfully used S19 (with formulations passing quality controls, an issue suspected in works included in these meta-analyses) [35,136]. As indicated above, controlled experiments are the first filter and the sustained field experience of countries where eradication succeeds is, as discussed below, invaluable.

In 2013, Denisov et al. [147] described that over 50 animal brucellosis vaccine strains were evaluated in the Russian Federation by 1996 by methods that are not described and provided a list of 22 accepted for veterinary practice; nevertheless, few came into use. These are *B. abortus* 104-M, *B. abortus* 82 (also known as SR82 [148]), *B. abortus* 75/79-AB and *B. abortus* KB 17/10, which, together with *B. abortus* 19 (B19 or S19) and *B. melitensis* Rev 1 (and possibly *B. abortus* 45/20 and *B. melitensis* 53Н38), complement the arsenal of vaccines that are probably used in this immense transcontinental Federation until (at least) 2013 [147,149,150]. There is no information on the effects of the vaccine 104-M (surprisingly used as a human vaccine in China [151]), and the same authors report that the vaccination of cattle was fairly evenly distributed between the live 82 and 75/79-AB vaccines. They also assert that using the former had significantly reduced cattle brucellosis, even though it showed abortifacient properties and culture instability [147,148]. Others also claim that vaccine 82 led to the eradication of cattle brucellosis in many areas of Russia [150]. Ivanov et al. [148] summarize the apparently favorable experiences with this vaccine, including outlines of controlled experiments, but do not provide the critical experimental details to obtain an independent opinion (Table 2). According to Shumilov et al. [149], the use of the live 75/79-AB vaccine led to the eradication of brucellosis in infected villages in less than 8 months, and the inactivated adjuvant KB 17/100 vaccine effectiveness “was evident in the rapid recovery of cattle brucellosis-infected farms within 6–24 months”. In sharp contrast with the experience of countries that eradicated ruminant brucellosis, S19 and Rev 1 (the latter unwisely used in both cattle and small ruminants), respectively, produced insufficient protection or unsatisfactory results in the field [147]. Considering that this methodology is not accessible or adequately described, that such effective results have no parallel anywhere, and that human brucellosis was still a severe problem by 2014 [147,152], all these results await confirmation, and the OIE does not endorse any of these Russian vaccines.

China, covering another extensive area of the world, also developed/used a repertoire of animal brucellosis vaccines that includes *B. abortus* A19, *B. melitensis* M5, *B. melitensis* M5-90, and *B. suis* S2. However, publications in international journals [151,153] only contain general claims on the extraordinary usefulness of these vaccines, which are not supported by well-documented experiments proving their safety and protection under controlled conditions (Table 2). Considering the dimensions of the claims, if such studies exist, they should have been made known with all the necessary methodological details, but no recent review on brucellosis in China has provided this information. Moreover, claims such as the great efficacy of S2 in all domestic livestock species when given orally would need robust and convincing experimental evidence. In fact, *B. suis* S2 is the only vaccine tested independently under controlled conditions (only against either *B. melitensis* or *B. ovis* in sheep), but the studies prove that it has only moderate to poor protective efficacy [154,155]. Reference research groups and institutions [156] recommended that this vaccine should not be used. Indeed, considering the current incidence of human brucellosis in China [16], the efficacy of all these vaccines is highly questionable and, like Russian vaccines, the OIE does not recommend any of these Chinese vaccines.

In contrast with the vaccines that are summarized above, a body of evidence shows that the live attenuated *B. abortus* S19 and *B. melitensis* Rev 1 vaccines developed in the first half of the 20th century have been decisive wherever the eradication of brucellosis in domestic ruminants has been attained, an achievement that was not accomplished by any program using other vaccines. The S19 and Rev 1 properties, the context and conditions of use, ancillary measures, and drawbacks have been reviewed, and the reader is directed to a recent FAO-sponsored publication for specific details [35].

S19 protects cattle against brucellosis caused by *B. abortus* or *B. melitensis* and was instrumental in eradicating bovine brucellosis in the United States, Australia, Canada, New Zealand, Scandinavia and several other European countries. No existing brucellosis vaccine is perfect, but the main drawbacks of S19 (i.e., abortions when vaccinating pregnant cows, induction of anti-S-LPS antibodies, and a low risk of infection in humans) were minimized or practically abrogated by using a conjunctively administered specific dose and appropriate personal protective equipment. Moreover, there are standardized procedures for vaccine production and, in contrast with other cattle vaccines (including RB51), well-established quality control protocols that, when implemented (which is not always the case), certify the stability and biological properties of S19 formulations. Despite its success, a tenacious effort to abandon S19 has persisted in the last three decades, to the point that it is not currently available in many countries. The reasons for this lie in the apparition of RB51, a continuous insistence on S19’s drawbacks, most specifically the induction of anti-S-LPS antibodies, and misunderstandings regarding the use of brucellosis vaccines, as discussed in Section 3.3.

Attempts were made many years ago to develop vaccines that solve some of the S19 drawbacks, such as the killed R *B. abortus* 45/20. These were soon abandoned due to their comparatively low efficacy, instability and the need for repeated immunization and untoward adjuvant-associated effects [35,157]. Then, in 1991, the R *B. abortus* RB51 live attenuated vaccine was developed, following the rationale that, as an R-mutant, it does not elicit anti-S-LPS antibodies and it would not interfere in serodiagnostic tests [137], which is misleading (see below)]. Initially, the RB51 vaccine for cattle was introduced in the United States in 1996 as a conditional vaccine, but only once bovine brucellosis was virtually eradicated, and the USDA banned S19 to avoid any diagnostic confusion [105]. Other high-income countries also soon reached eradication, which immediately contracted the “new” RB51 vaccine market. Then, RB51 (which, if efficacious (see below) would be helpful only in countries with very low herd prevalence using advanced T/S eradication programs) was introduced into resource-limited countries with severe deficiencies in their control programs including those in which T/S was never implemented [35,158]. However, after 30 years, no country using RB51 as the exclusive or primary vaccine has eradicated bovine brucellosis. Even countries with sufficient infrastructure, such as Chile and Uruguay, which had low brucellosis prevalence when RB51 was introduced and discontinued S19 many years ago, remain infected with sporadic outbreaks in both livestock and humans [159,160]. Why is this conundrum even possible? Part of the explanation lies in the plain fact, first suspected over fifteen years ago [137], and then fully established that RB51 does not solve any of the S19 drawbacks and introduces additional problems.

By the end of the 20th century, some investigators had reported that RB51 conferred the same, or even better, protection than S19. However, these claims were refuted in a subsequent evaluation of the corresponding experiments [35,137]. When appropriately examined, protection by RB51 is considerably lower than by S19 [137]. This asseveration is confirmed by the fact that brucellosis remains fully active in places where RB51 is administered, even when combined with T/S programs, with no signs of lowered prevalence and, in some cases, an aggravation of the brucellosis problem [35,70,101,103]. Indeed, the consequences of using a bad vaccine are a false sense of security and the subsequent relaxation of preventive management measures, factors that contribute to the perpetuation of brucellosis [161].

After the unsatisfactory results obtained with RB51 became apparent, re-vaccination with RB51 of cattle, first immunized with S19 or RB51, has become a widespread practice in Latin American countries, despite the absence of any proof of its safety and efficacy [35]. Nevertheless, in the mid-20th century, investigators demonstrated that revaccination against brucellosis, either with S or R vaccines, does not bring additional advantages; instead, it generates a complex combination of serological responses, complicating serodiagnosis [157,162]. Thus, it is not surprising that RB51 revaccination does not confer adequate protection under field conditions, aggravating the diagnostic interferences (see below) and increasing the false sense of security [35,70,161,163,164].

As indicated above, the rationale for developing RB51 was to avoid triggering antibodies against the S-LPS, which, if true, would make this strain a “DIVA” vaccine. Nevertheless, some countries that apply RB51 vaccination continue to use RBT and iELISAs or cELISAs (or FPA) as “screening/confirmatory” tests, respectively [70,101,165]. However, under those circumstances, no RBT-positive reaction can be “confirmed” (Section 3.4). Additionally, the R-LPS of RB51 elicits antibodies to the LPS lipid A-core epitopes that are detected in iELISAs, cELISAs, FPA and LFA but not in RBT, for topological reasons [48]. Therefore, in contrast to RBT, the DSp of these tests (the otherwise spurious reason for choosing them as “confirmatory” (Section 3.5)) considerably diminishes because a proportion of RB51-vaccinated (and revaccinated) animals become positive for iELISAs (in milk and serum), cELISAs, FPA and LFA [35,92,114], creating confusion. Moreover, antigenic stimuli caused by contact with field brucellae are common wherever vaccinated and infected animals coexist. Consequently, anti-S-LPS antibodies can be produced by all animals, whether or not they were vaccinated with RB51, and their sera will react as positive in all S-LPS tests, primary binding assays included. Therefore, even if they were protected, these animals would be classified as positive and culled in T/S programs [48]. This fact is a critical drawback of RB51 and any other R vaccine, making their otherwise imperfect “DIVA” properties useless and a surreptitious source of misdiagnosis when applied in infected environments. In these environments, the most sensible decision is to use the vaccine that affords the best protection, and this is not RB51. Unfortunately, most endemic countries have wholly disregarded these facts.

Twenty years ago, some authors also claimed that RB51 is safe in young and adult cattle, including pregnant ones [166,167,168,169,170]. Since then, it has been repeatedly confirmed that RB51 induces placentitis, endometritis, fetal colonization, vaginal excretion, mammary infections, milk excretion, and abortions [133,171,172,173,174,175,176], and both manufacturers and the OIE [136] disapprove of RB51 vaccination of pregnant cows. Still, despite the adverse effects recorded, various countries accept RB51 vaccination (and revaccination) of pregnant cattle. In addition, RB51 can infect humans [76], aggravating things even more. While S19 human infections can be diagnosed by simple serological tests and treated with conventional antibiotic regimes, the lack of S-LPS and rifampin resistance of RB51 precludes serodiagnoses with the standard tests and ambulatory antibiotic treatment (see Section 2.3). Following this, it is intriguing why and how RB51 became authorized in many countries. Although the antecedents are unclear, the consequences are not: introducing the RB51 vaccine in resource-limited countries has not alleviated the burden of brucellosis and has created confusion and distrust towards the programs and animal health authorities [133].

The cattle vaccine maze includes at least 11 cattle vaccines (possibly with more to come) that are claimed to be useful, but Ariadne’s thread (i.e., proven efficacy) has shown that *B. abortus* S19 is, at present, the only vaccine that can lead the way out of the labyrinth of cattle brucellosis.

## 4. Concluding Remarks

The literature on *Brucella* and brucellosis has rapidly increased over the last two decades and may continue to rise at the same pace, with the otherwise fortunate incorporation of scientists and professionals from endemic countries. During these 20 years, contributions concerning the structure, metabolism, genetics, pathogenesis, immunology, diversity and evolution of the *Brucella* genus have increased understanding in this field. Others, which may be relevant under specific contexts, reproduce previous experiences and confirm findings. However, many studies have little connection with current and past qualified research and the actual disease, and the real difficulty at present is in the ability to “separate the wheat from the chaff”. As shown here, the body of knowledge generated during the last three-quarters of the last century (accessible in journals, books, and some databases) is commonly overlooked or underrated in light of new technological procedures that, although valuable in some defined contexts, do not necessarily facilitate diagnosis, prophylaxis, and treatment in hospitals and diagnostic laboratories worldwide. This conundrum is particularly relevant in resource-limited countries, and they often bring confusion and further complicates economic and public health problems in these nations. Worst of all are decisions on the control of animal brucellosis that directly transpose the experiences and concepts that high-income countries used to fight brucellosis. Understanding the context in which this disease occurs is indispensable.

Believing that “old is always better than the new” is a romantic conviction, not a scientific attitude. However, the chances of learning anything valuable would be small if it was assumed that the new, expensive, and fancy is always better than the old, cheap, and simple. As the Spanish-American philosopher George Santayana wrote in The Life of Reason: “*Progress, far from consisting in change, depends on retentiveness…those who cannot remember the past are condemned to repeat it*” (Santayana 1950).

## Figures and Tables

**Figure 1 microorganisms-10-00942-f001:**
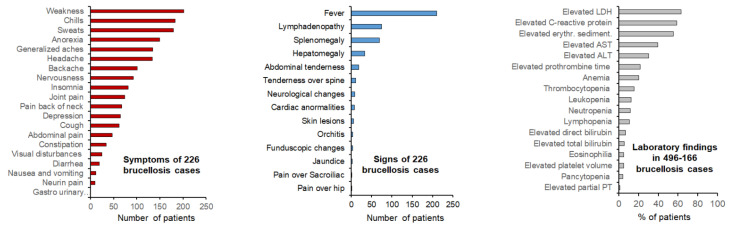
Diversity of clinical symptoms and laboratory findings in human brucellosis. Numbers of symptoms and signs in 250 brucellosis cases (mainly *B. abortus* infections) from the United States, as reported by Spink [23]. For laboratory findings, the percentages correspond to the total number (between 496 and 166) of patients (mainly *B. melitensis* infections) examined in a given test*,* as reported by Parlak et al. [8].

**Figure 2 microorganisms-10-00942-f002:**
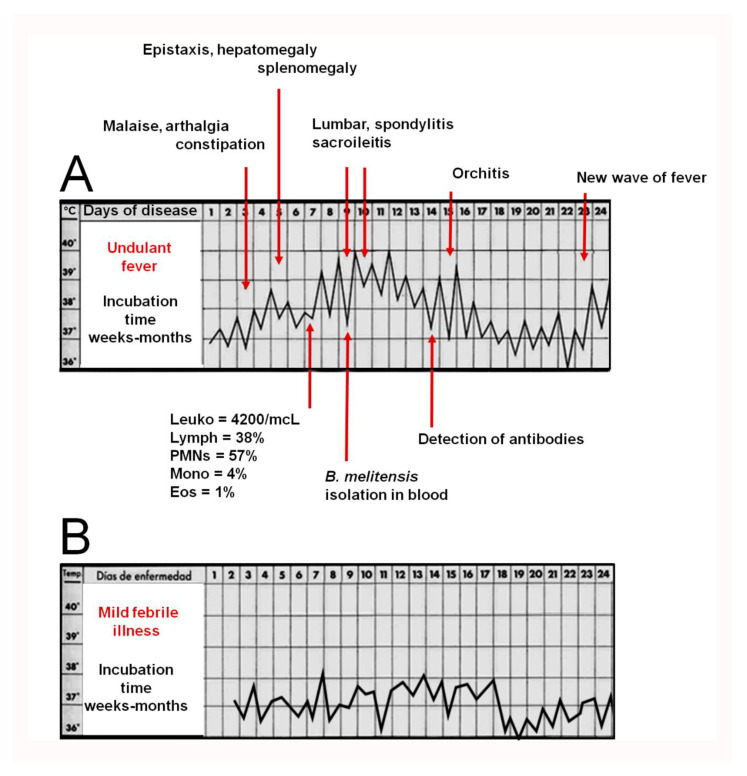
Clinical charts of two human brucellosis cases. (**A**) clinical chart displaying the “undulant fever” temperature of a brucellosis patient with subsequent clinical symptoms and signs. (**B**) Clinical chart of a brucellosis patient displaying mild febrile illness. Adapted from Reference [9].

**Figure 3 microorganisms-10-00942-f003:**
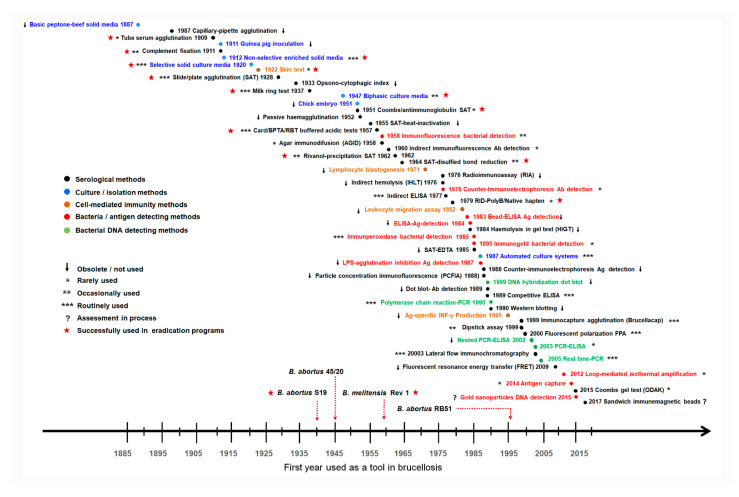
Diversity of brucellosis diagnostic tests developed over 130 years after the isolation of *B. melitensis* (*Micrococcus melitensis*) by David Bruce in 1887. The colored dots represent the time when the test was first used to diagnose brucellosis in animals, humans, or both. Many variant assays that use the same principle as the original method are not depicted (for instance, “non-selective culture media” includes many variants composed of blood, infusions or broths from plants, organs, yeast extracts, etc., with or without agar or gelatin). Likewise, the various selective media may include different antibiotics, colorants, and inhibitors. The “tube agglutination” method has several variants, including the centrifugation alternative and micro agglutination in 96-well round-bottomed plates. The antigens used in different tests (skin test, agar-gel immunodiffusion (AGID) and radial immunodiffusion (RID), counter-immunoelectrophoresis, INF-γ detection, Western blotting, blastogenesis, leukocyte migration, and all classical and binding assays) range from killed bacteria and crude bacterial extracts to enriched or purified preparations of proteins, LPS, core-O-PS, native hapten, polysaccharide B, and recombinant proteins. The primary binding methods use protein A, protein G, and polyclonal and monoclonal antibodies against LPS/O-PS epitopes or proteins as linking reagents. PCR methods use a wide range of primers based on sequences specific to the genus or a given species. Some agglutination assays use fixed bacteria or particles (e.g., erythrocytes or latex bead) coated with S-LPS or S-LPS hydrolytic polysaccharides, respectively, detecting antibodies against core-lipid A and O-PS or core-O-PS epitopes. We have included only the three vaccines that are extensively used. For other vaccines, see text.

**Table 1 microorganisms-10-00942-t001:** The fifty names of brucellosis *.

Adeno-tifo fever	Intermittent typhoid fever
Atypical infectious fever	Levant fever,
Atypical typhoid fever	Malta fever
Barcelona Fever	Mediterranean fever
Bruce septicemia	Mediterranean gastric remittent fever
Capricious fever	Ilo-tifo to sudoral form
Cartagena fever	Mediterranean tuberculosis
Cesspool fever,	Melitensis septicemia
Climatic fever	Melitococcia
Continuous epidemic fever	Melitosis
Corps disease	Mephitic fever,
Country fever,	Miliary fever
Crazy fever	Napolitan Fever
Cretan fever,	One 100 clinical form disease
Crimean fever	Phthisis
Cyprus fever,	Pseudo-tifo
Dust fever	Pythogenic septicemia
Faeco-malarial fever	Recurrent fever
Febricola typhosa	Remittent fever
Febris complicata	Rock fever
Febris *melitensis*	Sewage fever,
Febris sudoralis	Simple continued fever
Gastro-bilious fever	Town fever,
Gibraltar fever	Typho-malarial fever
Goat fever	Undulant fever

* The names and the appropriate references are cited in [2,5,20,25].

**Table 2 microorganisms-10-00942-t002:** Requisites for valid controlled experiments of brucellosis vaccines in target hosts.

Requisite		Comments
Animals	Brucellosis-free status	Any contact with the pathogen would biased results in impossible to predict and individually variable ways. In areas not free of brucellae, selection based on serology is not acceptable because of the test(s) DSe/DSp and latent carrier issues, particularly in young animals.
	Other diseases	Healthy and rigorously proved to be free from confounding pathologies and infections, particularly those causing reproductive failure.
	Homogeneity	Breed, age, sex, and similar physiological status; pregnancy synchronized and monitored throughout the process.
Challenge	Route	Through the conjunctiva because it reproduces a common natural infection route (subcutaneous, intravenous, or intramuscular are highly artificial).
	Strain	In a previous test in the laboratory performing the experiments, the strain has to reproduce the multiplication in mouse spleens characteristic of virulent strains. Extreme precautions should be taken to avoid any degree of attenuation, including master seed/inoculum strategy and S to R dissociation controls for every inoculum.
	Vaccination-challenge interval	From 6 months to 1 year, depending on the age at vaccination (shorter periods do not provide information on sustained immunity).
	Pregnancy	Pregnant animals should be challenged when most susceptible to abortion (mid-pregnancy). Later times progressively reduce the development of lesions of a high enough intensity to damage fetus development.
Assessments	Selective media	Strictly necessary. Commensal bacteria easily overgrow brucellae in milk, vaginal fluids, or semen. It is technically unfeasible to obtain perfectly clean necropsy samples of organs, even after surface disinfection, and microorganisms other than brucellae can be present within lymph nodes. The selective medium (better a combination of media) should be chosen, noting that some are inhibitory for some species or strains (Farrell’s for *B. melitensis*, *B. ovis,* and *B. abortus* RB51).
	Detection threshold	As brucellae can be present in insufficient numbers and excreted and/or increase after sexual maturity or pregnancy, maximizing detection is critical. Thick homogenates in a minimal volume of diluent should be directly seeded (up to 0,5 mL of tissue/plate) on several plates (sensitivity up to 1 CFU/g of tissue), not only dilutions.
	Abortions	Protection against abortion with no complete protection is not valid as it can be counterproductive (non-aborted infected animals shed brucellae after delivery and give birth to infected offspring serologically negative latent carriers until first pregnancy when they abort and spread the disease).
Controls	Non vaccinated	Mandatory. The infection rate should be close to 100%, and the infecting strain is widespread among organs/lymph nodes.
	Reference vaccine	S19 and Rev 1 controls (OIE reference vaccines; CFU number and absence of dissociation assessed) are strictly necessary.

## Data Availability

Not applicable.

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
