# Peer review of "Facing the Human and Animal Brucellosis Conundrums: The Forgotten Lessons"

_microorganisms, 2022, doi:10.3390/microorganisms10050942_

Round 1
Reviewer 1 Report
- Please, add line numbers;
- Micrococcus melitensis (Brucella melitensis): which is the right name? Please, clarify.
- Aims and predictions are unclear and needs some better explanations.
- Figure 1 is unreadable, please rewrite axis-labels.
- Table 1. “The list of names and the appropriate references are cited in (Evans, 1918; Hughes, 1896; Ruiz-Castañeda, 1986; Spink, 1956c”. Please, follow the reference style asked by the journal.
- Figure 2. Why you used italics? Please, cite the reference following the MDPI style.
- “A non-exhaustive PubMed search (Supplemental material PCR studies) reveals over 20 different PCR amplification protocols tested under laboratory conditions”. Carry out an exhaustive search. This would be really important in a research study like this one.
- Throughout the text, tables and figures show different fonts. Please be consistent.
Author Response
Please, see the attachment

Reviewer 2 Report
The authors write a beautiful review about the history of brucellosis and the difficulty of diagnosing a disease with a wide variety of symptoms. the review is well written and every aspect is addressed in a complete and exhaustive way. It was a pleasure to read itI just have a few small tips to give:
1: Abstract its 203 words and not 200.
2: Figure 1:Why do you report symptoms and signs as number of patients, while the laboratory findings as a percentage of patients? It would be better if all three objects analyzed are reported in the same way (i.e. number of patients)
Besides, it’s not clear what 496-166 are. Please, specify
3: Please, Replace Footnote 1 with Footnote 2 and vice versa
4: Please, delete “shared by the editor and reviewers of this paper” Pag.9 after reference 44
5: Please, delete Bibliography}
6: the references format in supplementary materials must be arranged according to the guidelines of the journal
Author Response
Please, see the attachment.

Reviewer 3 Report
Its a good start for a review in the field of Brucella research. However, the nomenclature review is not completely as it should not only be focus on the different names of diseases but also the chaos of the taxonomy itself - until the latest assignment of Ochrobactrum species to the Brucella species. However, this is only a minor point. A main issue is that about 90% of the cited literature was considered by the authors to be wrong. I totally agree that there is much research published which is methodological not correct or in which wrong conclusions are drawn. Such research must be "politically correct" compared to correct approaches. Terms that can be used are "in contrast to" or "discrepancies between the study A and B...". Ethics in sciences prohibits valuation of studies with simple defamations as for example "without information" or "useless". A scientific based comparison of studies (not only your opinion) is mandatory and if you are successful with this the reader will draw the conclusion self. There are hundreds of brilliant researchers in the field of brucella but you only mention (in your eyes) negative and wrong studies and ignore the positive research.

Author Response
Please, see the attachment.

Round 2
Reviewer 1 Report
Authors addressed most of my previous comments.
1. However, the MS is still a bit confused, some words are marked in yellow, other corrections are red.
2. Still, aims and predictions have not ben corrected, but if the Editor is ok, I am ok too.
3. Still, I do not understand why you used italics in some figures.
All the best
Author Response
Answer reviewer 1 Round 2
Again, many thanks for reviewing our manuscript.
1. However, the MS is still a bit confused, some words are marked in yellow,
other corrections are red.
We regret the confusion and apologize for this. The red words represented
modifications. The words marked in yellow were left-overs of documents exchanged between the authors for discussions on modifications that we missed to remove in the revised manuscript. In the last version, they have been removed.
2. Still, aims and predictions have not been corrected, but if the Editor is ok, I
am ok too.
There must be (again) a problem with the version of the manuscript that reached you because, indeed, we followed your advice and modified the aims and predictions (lines 105-122 of modified manuscript; in red characters).
3. Still, I do not understand why you used italics in some figures.
In our previous answer, we indicated that the idea was to differentiate some texts for printing in grey tones. However, since this explanation does not satisfy the reviewer, we have modified Figure 2 and used normal instead of italic characters (see revised Figures).

Reviewer 2 Report
I thank authors for replying to all my comments. I support the publication of the manuscriptAuthor Response
Attached

Reviewer 3 Report
no comments
